# M-Color: MLLM-Guided Diffusion Models for Image Colorization

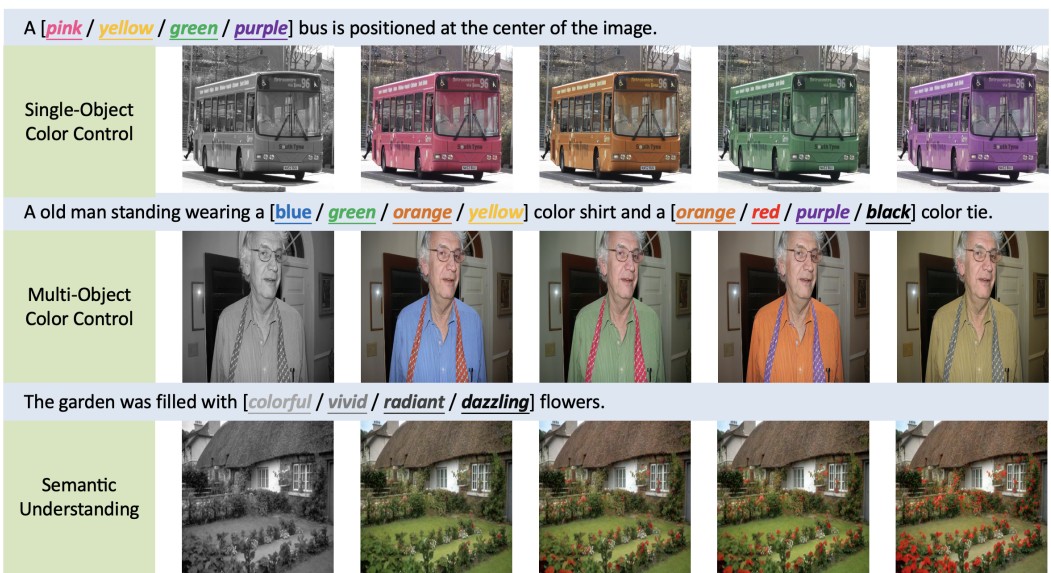

Figure 1: M-Color results across three scenarios: single-object color control (first row, bus), multi-object color control (second row, clothing), and semantic understanding (last row, colorfulness).

## ABSTRACT

Language-based Image colorization transforms grayscale images into vivid, visually pleasing colorized outputs with semantic guidance. Existing methods often rely on CLIP text embeddings, which may struggle with deep semantic understanding, leading to suboptimal colorization. In this paper, we propose M-Color, a novel diffusion-based framework that leverages multimodal large language models (MLLMs) to enhance language comprehension through an Adaptive Decoding strategy. To maintain structural consistency, we introduce a Luminance-Aware Encoder (LAE) that aligns grayscale images with the colorized output and a Luminance Extraction Module (LEM) to integrate luminance information into the latent generation process. Extensive experiments demonstrate that M-Color achieves superior semantic alignment, improves structural consistency, and outperforms state-of-the-art methods in both quantitative and qualitative evaluations.

**Keywords**: language-based image colorization, diffusion models

[1]

## 1 INTRODUCTION

Image colorization transforms grayscale images into vivid, semantically meaningful versions and finds applications in historical photo restoration and creative media production. However, traditional

---

[1]ChatGPT was used for minor writing assistance. It generated no technical content or ideas.

automatic methods Cheng et al. (2015); Vitoria et al. (2020); Su et al. (2020); Weng et al. (2022a); Cong et al. (2024) often suffer from color ambiguity and lack of user control, leading to unnatural color assignments. To improve controllability, user-guided methods have been proposed, such as reference-based Lu et al. (2020); Bai et al. (2022), scribble-based Zhang et al. (2017); Yun et al. (2023), and palette-based Wang et al. (2022) approaches. While these offer varying degrees of guidance, they still fall short in achieving fine-grained, semantically aligned colorization.

Language-based colorization Weng et al. (2022b); Chang et al. (2022; 2023); Zabari et al. (2023); Chang et al. (2024); Li et al. (2024) has emerged as an intuitive solution using natural language prompts. Yet, prior works relying on CLIP Radford et al. (2021) or BERT Devlin et al. (2019) face challenges in deep semantic understanding and text-image alignment. Moreover, maintaining structural consistency remains difficult, especially in diffusion-based methods that lack proper integration of grayscale structures.

To overcome these limitations, we propose M-Color, an MLLM-guided diffusion framework for language-based image colorization. As shown in Figure 1, M-Color excels in single-object and multi-object color control and captures semantic nuances such as varying levels of vividness, demonstrating its strong alignment with language prompts. Our approach builds on Stable Diffusion Rombach et al. (2022) and leverages Multi-modal Large Language Models (MLLMs) for richer text-image alignment. To enhance guidance and consistency, we introduce three key components: Adaptive Decoding for semantically grounded text embedding, a Luminance-Aware Encoder (LAE) for preserving structural detail, and a Luminance Extraction Module (LEM) to inject luminance cues into the latent space, mitigating color overflow and distortion.

## 2 RELATED WORK

### 2.1 LANGUAGE-BASED IMAGE COLORIZATION

Early language-based colorization methods relied on LSTMs (e.g., L-CoDe Weng et al. (2022b)) or transformer models like BERT (e.g., L-CoDer Chang et al. (2022), L-CoIns Chang et al. (2023)), but often lacked fine-grained alignment between text and image regions. Recent methods such as UniColor Huang et al. (2022), L-CAD Chang et al. (2024), and COCO-LC Li et al. (2024) utilize CLIP Radford et al. (2021) for stronger multimodal representations. However, CLIP is not tailored for colorization and may fail to capture complex scene semantics, leading to color mismatches. We address this by introducing MLLMs to provide richer semantic understanding and better alignment, improving both color accuracy and user control.

### 2.2 DIFFUSION-BASED IMAGE COLORIZATION

Diffusion models have shown strong generative capabilities in various tasks Lugmayr et al. (2022); Brooks et al. (2023); Fu et al. (2024); Gao1 et al. (2023), making them well-suited for colorization. L-CAD Chang et al. (2024) introduces luminance-guided compression for structural preservation, while Diffusing Colors Zabari et al. (2023) applies cold diffusion Bansal et al. (2023) to improve stability. CtrlColor Liang et al. (2024) builds on ControlNet Zhang et al. (2023) for enhanced spatial control, and COCO-LC Li et al. (2024) allows user-adjustable saturation. These approaches improve realism and control, yet challenges remain in jointly preserving structure and following textual guidance—gaps our method aims to close.

### 2.3 MLLM-GUIDED IMAGE GENERATION

MLLMs like LLaVA Liu et al. (2023) offer vision-language understanding beyond pre-trained encoders like CLIP, enabling better control in generation tasks. Works such as GILL Koh et al. (2023), MGIE Fu et al. (2024), and SmartEdit Huang et al. (2024) use MLLMs for text-conditioned image generation and editing, demonstrating enhanced text-image alignment and interpretability. Unlike image editing, colorization requires global semantic transformation of grayscale inputs. Editing models often lack the contextual color priors necessary for coherent colorization, motivating our MLLM-guided diffusion framework to bridge this gap.

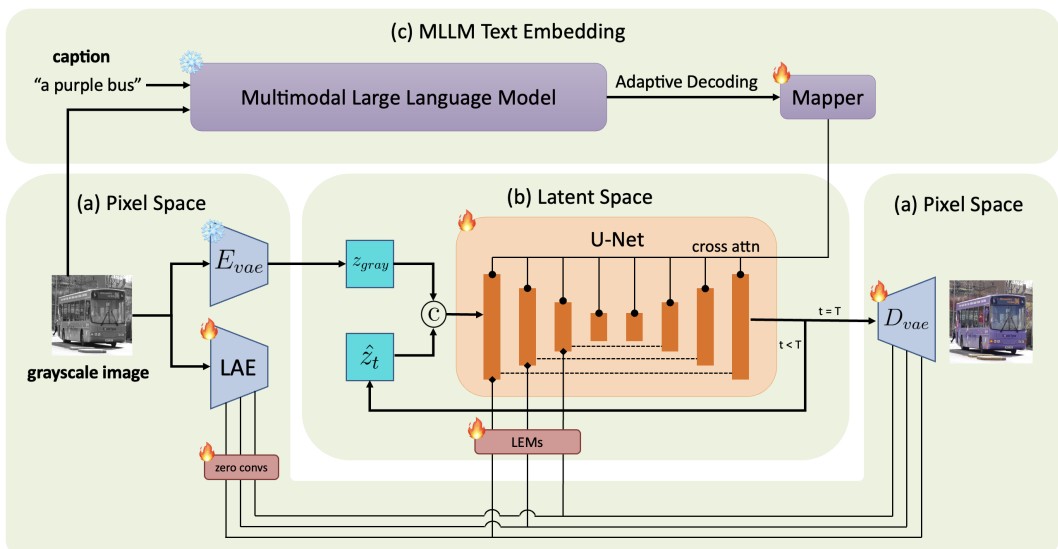

Figure 2: The overall architecture of M-Color, consists of (a) Pixel Space, (b) Latent Space, and (c) MLLM Text Embedding.

## 3 PROPOSED METHOD

We present M-Color, an MLLM-guided diffusion-based colorization framework. As shown in Figure 2, the overall architecture consists of three parts: (a) Pixel Space, (b) Latent Space, and (c) MLLM Text Embedding, detailed in Sections 3.2, 3.3, and 3.4, respectively. Section 3.1 reviews Stable Diffusion and VAE-induced distortions. Section 3.2 presents Luminance-Aware Image Reconstruction for structural consistency. Section 3.3 introduces Dual-Encoder Latent Generation to separate semantic and luminance features. Section 3.4 proposes MLLM-guided Text Embedding with Adaptive Decoding for enhanced semantic-text alignment and vivid colorization.

### 3.1 PRELIMINARY: STABLE DIFFUSION

Stable Diffusion Rombach et al. (2022) is a latent diffusion model (LDM) that generates high-quality images by iteratively denoising latent representations, rather than working directly in pixel space. It introduces Gaussian noise to a latent $x_t$ over time and learns to reverse this process by predicting the noise $\epsilon$ using a neural network $\epsilon_\theta(x_t, t)$, optimized by minimizing $\mathbb{E}_{x_0,t,\epsilon}[\|\epsilon - \epsilon_\theta(x_t, t)\|^2]$. The latent representations are produced by a Variational Autoencoder (VAE), where an image $x$ is encoded as $z = E(x)$ and reconstructed as $\hat{x} = D(z)$. While operating in latent space improves efficiency, the VAE introduces structural distortion, motivating the development of modules to better preserve fine details during reconstruction.

### 3.2 LUMINANCE-AWARE IMAGE RECONSTRUCTION

The Variational Autoencoder (VAE) used in Stable Diffusion often introduces structural distortions due to its lossy compression and reconstruction process. These distortions result in a misalignment between the reconstructed output and the original grayscale input, impairing the structural consistency needed for accurate colorization.

To address this issue, previous work such as L-CAD Chang et al. (2024) proposed a dual-encoder framework that supplements the standard VAE encoder with a luminance-specific encoder to preserve structural details. Inspired by this idea, we also adopt a dual-encoder design, introducing a separate Luminance-Aware Encoder (LAE) alongside the standard encoder $E_{vae}$, which specifically extracts luminance information from the grayscale input $x_{gray}$.

Our method differs from L-CAD in two key aspects: (1) we adopt a ControlNet-like Zhang et al. (2023) design, injecting LAE features into the decoder $D_{vae}$ through residual connections with zero

convolution layers, which modulate the feature influence without introducing bias; (2) we jointly train the LAE and VAE decoder to ensure better convergence and effective luminance preservation during reconstruction.

We optimize this stage using a combination of reconstruction and perceptual loss, formulated as:

$$\mathcal{L}_{pixel} = \alpha \cdot \|x - \hat{x}\|_2 + \beta \cdot \text{vgg\_loss}(x, \hat{x}), \tag{1}$$

where $\alpha = 1$ and $\beta = 0.1$ balance pixel-level accuracy and perceptual quality.

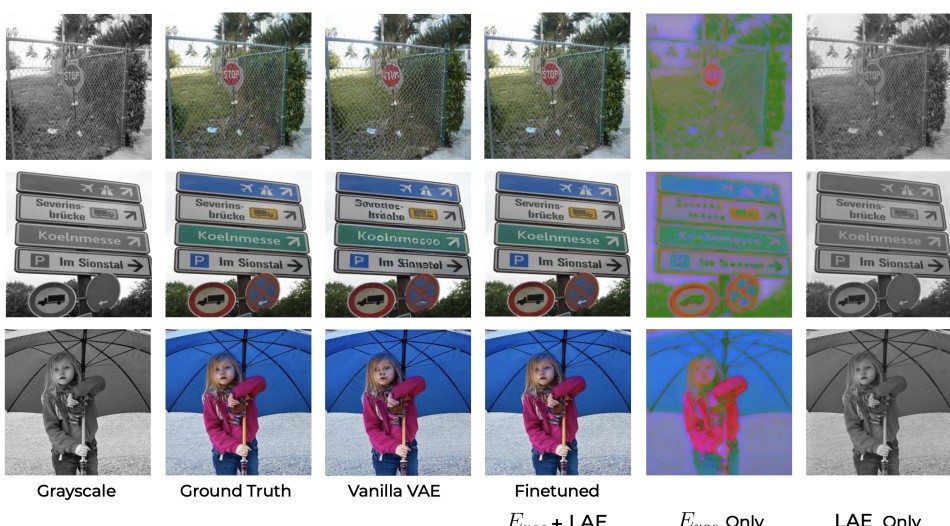

Figure 3: Results of Luminance-Aware Image Reconstruction

Figure 3 shows the results of Luminance-Aware Image Reconstruction. The first two rows present the grayscale inputs and ground truth colors, while the third and fourth rows show reconstructions from the vanilla VAE and the fine-tuned VAE with LAE. The vanilla VAE introduces distortions in faces, text, and dense details, whereas the fine-tuned VAE with LAE resolves these issues and improves structural fidelity. To analyze encoder roles, we tested $E_{\text{vae}}$ and LAE separately: $E_{\text{vae}}$ captures semantic content, while LAE preserves luminance and fine structures. Their combination is essential for high-quality reconstruction and will be leveraged in later sections.

### 3.3 DUAL-ENCODER LATENT GENERATION

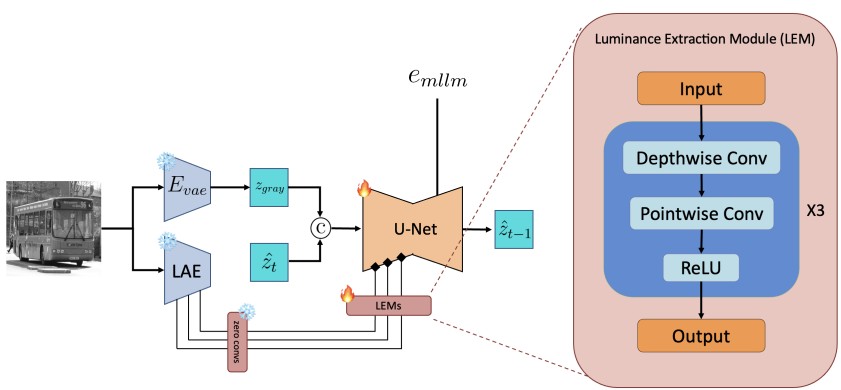

Figure 4: Architecture of Dual-Encoder Latent Generation

In the previous section, we introduced the Luminance-Aware Encoder (LAE) to complement the VAE encoder ($E_{\text{vae}}$), enabling separate extraction of semantic and luminance information. This

dual-encoder design improves pixel-space reconstruction. Building on this, we extend the approach to latent generation (Figure 4). Unlike prior works Chang et al. (2024); Li et al. (2024); Liang et al. (2024) that rely on either semantic or luminance cues, our method integrates both for better alignment between text, structure, and color.

For semantics, we concatenate the grayscale latent from $E_{\text{vae}}$ with the noise at each diffusion timestep and feed it into the U-Net, providing content-aware guidance. For luminance, we inject LAE features via residual connections, similar to Section 3.2. Since LAE features are high-resolution while the U-Net operates on compressed latents, we introduce the Luminance Extraction Module (LEM). LEM uses depthwise and pointwise convolutions to adapt LAE outputs to the U-Net's spatial and channel dimensions, enabling effective luminance integration while preserving alignment and consistency.

To train the latent generation, we adopt an $\ell_2$ loss under the standard diffusion noise prediction formulation:

$$\mathcal{L}_{\text{latent}} = \mathbb{E}_{\epsilon \sim \mathcal{N}(0,I),\ t,\ z_0} \left[ \|\epsilon_t - \epsilon_\theta(z_t,\ z_{\text{gray}},\ r_{\text{lum}},\ e_{\text{mllm}},\ t)\|^2 \right]. \tag{2}$$

Here, $z_t$ is the noisy latent, $z_{\text{gray}}$ the grayscale latent from $E_{\text{vae}}$, $r_{\text{lum}}$ the luminance residual from LAE via LEM, and $e_{\text{mllm}}$ the language embedding. The denoiser $\epsilon_\theta$ conditions on all four to exploit both semantic and structural cues.

For controllable text guidance, we apply Classifier-Free Guidance Ho & Salimans (2021) on $e_{\text{mllm}}$. During training, $e_{\text{mllm}}$ is replaced with a null embedding $\phi$ with 10% probability. At inference, outputs interpolate between conditional and unconditional predictions with guidance scale $s = 3.5$:

$$\begin{aligned} \epsilon_{\text{cfg}} = s \cdot \epsilon_\theta(z_t,\ z_{\text{gray}},\ r_{\text{lum}},\ e_{\text{mllm}},\ t) \\ + (1 - s) \cdot \epsilon_\theta(z_t,\ z_{\text{gray}},\ r_{\text{lum}},\ \phi,\ t). \end{aligned} \tag{3}$$

## 3.4 MLLM-GUIDED TEXT EMBEDDING

---
**Algorithm 1** Adaptive Decoding

---
**Require:** Prompt $p$, Grayscale image $I$, Max decoding steps $T$
**Ensure:** Token sequence $Y = \{y_1, y_2, \ldots, y_n\}$
1: Initialize $Y \leftarrow [\,]$, $P_{[\texttt{IMG}_0]} \leftarrow [\,]$
2: **for** $t = 1$ to $T$ **do**
3:     Compute token probabilities $P(y_t \mid Y_{<t}, p, I)$
4:     Select token $y_t \leftarrow \arg\max_{y \neq [\texttt{IMG}]} P(y)$                ▷ ignore visual tokens
5:     Append $y_t$ to $Y$
6:     Append $P(y_t = [\texttt{IMG}_0])$ to $P_{[\texttt{IMG}_0]}$
7:     **if** $y_t = [\texttt{EOS}]$ **then**
8:         **break**
9:     **end if**
10: **end for**
11: $i^* \leftarrow \arg\max_t P_{[\texttt{IMG}_0]}$
12: $Y \leftarrow Y_{[:i^*]} \,\|\, [\texttt{IMG}_0] \ldots [\texttt{IMG}_{n-1}][\texttt{EOS}]$
13: **return** $Y$

---

Recent advances in multi-modal large language models (MLLMs) show strong capability in joint visual–text understanding Koh et al. (2023); Fu et al. (2024). We leverage MLLMs as a semantically richer alternative to CLIP-based embeddings in diffusion models. For colorization, we rephrase the image caption into an instruction prompt (*"Given the grayscale image, how would it look if it were colored like [CAP]"*) and feed it, along with the grayscale image, into a pretrained MLLM. The MLLM produces detailed scene descriptions and $n$ visual tokens $[\texttt{IMG}]$ that encode high-level semantics.

The hidden states of $[\texttt{IMG}]$ tokens are projected by a lightweight transformer into a visual guidance space for U-Net conditioning. We adopt MGIE Fu et al. (2024), but observe that it often emits $[\texttt{IMG}]$ prematurely, degrading guidance. To address this, we propose *Adaptive Decoding*, inspired by Skip Decoding Qian et al. (2024). Instead of emitting visual tokens immediately, we first decode text tokens, track the probability of $[\texttt{IMG}]$, and insert them retrospectively at the position with peak probability after sequence completion.

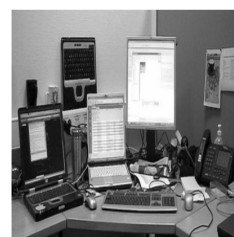

| Grayscale | Caption | MLLM Response |

A silver thermos. Computer monitor turned on. A white and pink coffee mug. A silver mouse by two laptops. A pad of paper.

The image would feature a workspace with two computers, one on the left and another on the right side of the laptop. There would be a desktop computer monitor on the left side, and a desktop computer mouse on the right side. Additionally, there would be two laptops, one on the left and another on the right. A silver tray, white coffee mug, and a silver mouse would also be present in the scene. Lastly, there would be a cell phone placed near the desktop computer mouse. Lastly, there would be a paper notebook placed near the desktop computer monitor. [IMG0][IMG1][IMG2][IMG3][IMG4][IMG5][IMG6][IMG7]

Figure 5: Example of Adaptive Decoding results

This strategy improves semantic-text alignment by placing visual tokens at optimal positions, enhancing downstream colorization guidance. Figure 5 illustrates the process, where red text marks the additional fragment and optimal insertion point. Here, visual tokens range from `[IMG0]` to `[IMG7]`.

# 4 EXPERIMENT

## 4.1 IMPLEMENTATION DETAILS

We conduct all experiments using four NVIDIA V100 GPUs with a training resolution of $256 \times 256$. Since we use a frozen MLLM as the source of text conditioning, we pre-generate the hidden states of the visual tokens prior to training to avoid redundant computation. In the pixel-space stage, we set the learning rate to $1 \times 10^{-4}$ and train the model for 10 epochs until convergence. For the latent-space stage, we first train with a learning rate of $1 \times 10^{-4}$ for 40 epochs, followed by an additional 10 epochs with a reduced learning rate of $1 \times 10^{-5}$ to further refine the generation quality. The entire training pipeline takes 60 hours to complete.

Similar to InstructPix2Pix Brooks et al. (2023), we train our model at a resolution of $256 \times 256$, but perform inference at $512 \times 512$. We find that this approach consistently improves the visual quality of the generated images, producing sharper details and more coherent colorization results. During inference, we use the PNDM scheduler Liu et al. (2022) for accelerated sampling and perform 20 denoising steps, which strikes a good balance between speed and quality.

## 4.2 DATASETS

We conduct experiments on two language-based colorization datasets: (1) the extended COCO-Stuff Weng et al. (2022b), obtained by filtering unsuitable samples, and (2) the multi-instance dataset Chang et al. (2023), which contains images with multiple distinct objects.

Following COCO-LC Li et al. (2024), we remove grayscale-only images. For the multi-instance dataset, we also found duplicates between training and validation sets, often with resolution differences. To mitigate this, we apply a CLIP-based embedding distance filter, removing about 20% of validation samples and reducing overfitting risk.

Each sample is an image–caption pair. After filtering, the cleaned COCO-Stuff* dataset contains 101K training and 4K validation pairs, while the multi-instance* dataset has 114K training and 10K validation pairs, where '*' denotes the cleaned versions.

## 4.3 COMPARISON

We compare our method with existing language-based colorization approaches (e.g., L-Code Weng et al. (2022b), L-CoDer Chang et al. (2022), L-CAD Chang et al. (2024)) and multi-modal colorization methods (e.g., UniColor Huang et al. (2022), CtrlColor Liang et al. (2024)) under textual conditioning.

Table 1: Quantitative results of comparison with state-of-the-art methods on MSCOCO and Multi Instance datasets. Best results are highlighted in red. * indicates modified dataset. + indicates diffusion-based methods.

| Extended MSCOCO * | | | | Multi Instance * | | | |
|---|---|---|---|---|---|---|---|
| Method | PSNR ↑ | SSIM ↑ | LPIPS ↓ | Method | PSNR ↑ | SSIM ↑ | LPIPS ↓ |
| UniColor | 21.85 | 0.8905 | 0.2325 | UniColor | 20.60 | 0.8801 | 0.2498 |
| CtrlColor + | 21.31 | 0.8679 | 0.2362 | CtrlColor + | 21.04 | 0.8657 | 0.2369 |
| L-CoDe | 22.59 | 0.7938 | 0.3605 | L-CoDe | 22.30 | 0.8056 | 0.3448 |
| L-CoDer | 22.79 | 0.7918 | 0.3568 | L-CoDer | 22.45 | 0.7936 | 0.3521 |
| L-CAD + | 23.42 | 0.9030 | 0.1885 | L-CAD + | 22.88 | 0.9028 | 0.1930 |
| **M-Color +** | **24.75** | **0.9209** | **0.1646** | **M-Color +** | **24.40** | **0.9213** | **0.1655** |

### 4.3.1 QUANTITATIVE RESULTS

Following prior work Weng et al. (2022b); Chang et al. (2022; 2023; 2024), we evaluate colorization quality using PSNR Huynh-Thu & Ghanbari (2008), SSIM Wang et al. (2004), and LPIPS Zhang et al. (2018). Since metrics are resolution-sensitive, all results are resized to 512×512, the largest resolution among compared methods. For diffusion-based approaches Chang et al. (2024); Liang et al. (2024), we adopt the same setup as M-Color, generating images at 512×512 with 20 sampling steps, ensuring fair evaluation.

As shown in Table 1, M-Color achieves the best performance on all three metrics across both datasets, confirming its superior semantic coherence and visual fidelity.

### 4.3.2 QUALITATIVE RESULTS

Previous works on language-based colorization often use small text encoders like CLIP, leading to incomplete semantic understanding. In addition, most methods rely solely on semantic or luminance cues during latent generation, causing color-object mismatches or overflow.

As shown in Figure 6, M-Color achieves the best alignment between prompts and colorized images. It improves color fidelity, mitigates overflow, and enhances visual quality. For example, in the third row, prior methods show overflow on the person's legs, while M-Color preserves boundaries and renders the wheel in orange. In the last row, M-Color also handles complex multi-object instructions with accurate and consistent colors.

### 4.4 USER STUDY

To evaluate our method, we conducted two user studies: one on text-image matching and another on image realism. In the first, participants selected the image that best matched a given caption; in the second, they chose the most realistic image from a given set. Each test included 20 randomly sampled questions, with 32 volunteers participating. As shown in Table 2, our method outperformed all other methods, achieving the highest scores in both tasks.

Table 2: Results of text matching and image realism user study.

| Text Matching Experiment | | Image Realism Experiment | |
|---|---|---|---|
| Method | Ratio | Method | Ratio |
| UniColor | 12.19% | UniColor | 5.33% |
| CtrlColor | 12.28% | CtrlColor | 9.53% |
| L-CoDer | 6.32% | L-CoDer | 10.18% |
| L-CAD | 31.21% | L-CAD | 13.27% |
| Ours | 38.01% | Ours | 22.90% |
| - | - | Groundtruth | 38.80% |

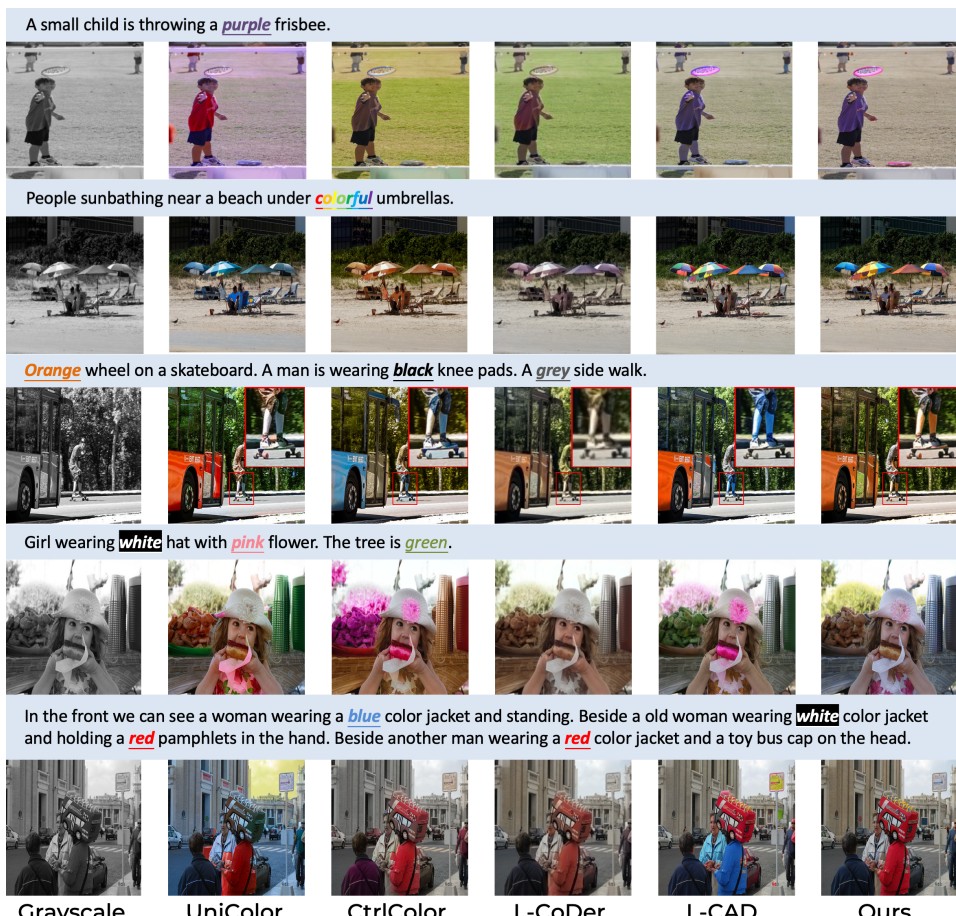

Figure 6: Comparison with state-of-the-art methods

## 4.5 ABLATION STUDY

To assess the contribution of each component in M-Color, we conduct five ablation experiments: (1) **clip text**, which replaces the MLLM-based text embedding with CLIP to examine the impact of a smaller language model; (2) **w/o AD**, which removes Adaptive Decoding and reverts to vanilla decoding; (3) **w/o LAE**, which removes the Luminance-Aware Encoder and uses the vanilla VAE without fine-tuning, thereby omitting structural consistency enhancements; (4) **w/o sem**, which excludes semantic cues by removing grayscale latent concatenation at each diffusion timestep; and (5) **w/o lum**, which removes luminance cues by disabling the Luminance Extraction Modules (LEMs) and residual connections into the U-Net.

### 4.5.1 QUANTITATIVE RESULTS

Quantitative results from these ablation experiments are shown in Table 3. It can be observed that the removal of each component leads to a performance drop to varying degrees, show the importance of each module in achieving optimal colorization quality.

### 4.5.2 QUALITATIVE RESULTS

Figure 7 presents qualitative results of the ablation study, revealing how the removal of specific components affects output quality. In the **clip text**, **w/o AD**, and **w/o sem** settings, color-object mismatches are common due to insufficient semantic guidance, leading to colors being applied to unintended regions.

Table 3: Quantitative results of ablation study on MSCOCO and Multi Instance datasets. Best results are highlighted in red. * indicates modified dataset.

| Extended MSCOCO * | | | | Multi Instance * | | | |
|---|---|---|---|---|---|---|---|
| Method | PSNR ↑ | SSIM ↑ | LPIPS ↓ | Method | PSNR ↑ | SSIM ↑ | LPIPS ↓ |
| clip text | 24.67 | 0.9196 | 0.1684 | clip text | 24.23 | 0.9194 | 0.1698 |
| w/o AD | 24.73 | 0.9208 | 0.1647 | w/o AD | 24.38 | 0.9213 | 0.1656 |
| w/o LAE | 21.81 | 0.7273 | 0.2672 | w/o LAE | 21.43 | 0.7258 | 0.2682 |
| w/o sem | 24.62 | 0.9187 | 0.1682 | w/o sem | 24.32 | 0.9202 | 0.1671 |
| w/o lum | 24.60 | 0.9180 | 0.1707 | w/o lum | 24.32 | 0.9201 | 0.1694 |
| **M-Color** | **24.75** | **0.9209** | **0.1646** | **M-Color** | **24.40** | **0.9213** | **0.1655** |

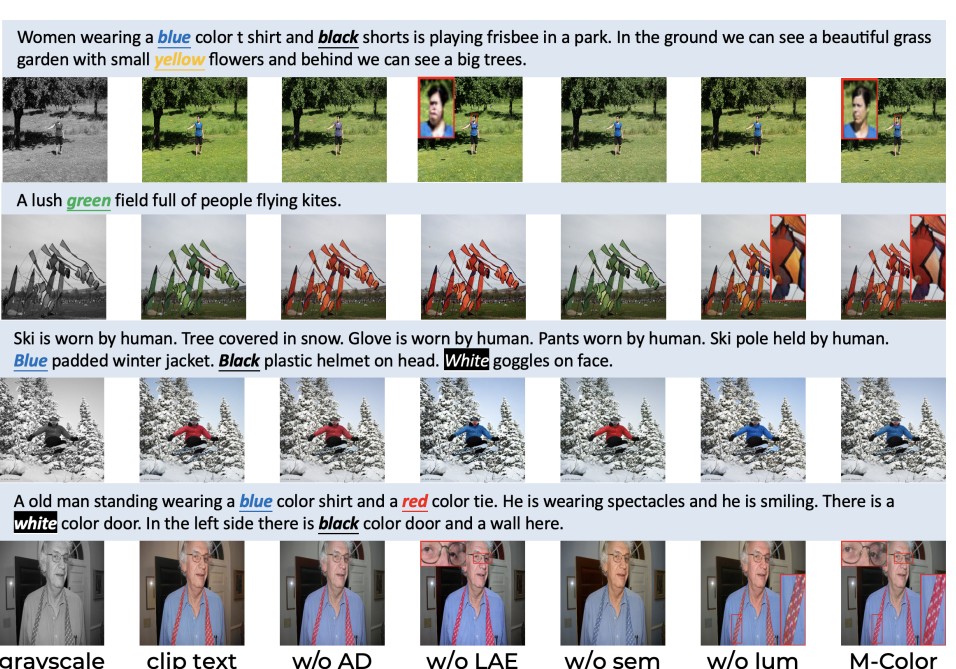

Figure 7: Qualitative results of ablation study

The **w/o LAE** setting causes noticeable structural distortions, especially in faces and fine details, since the model lacks structural preservation, as seen in distorted facial features and inconsistent eye regions.

In the **w/o lum** setting, removing luminance cues results in color overflow and imprecise boundaries. This is evident in saturated object edges (e.g., the second image) and incorrect coloring of localized regions (e.g., the tie in the fourth image).

## 5 CONCLUSION

We presented M-Color, a diffusion-based framework for language-guided image colorization that leverages Multi-modal Large Language Models (MLLMs) to enhance semantic-text alignment and structural consistency. Key modules including the Luminance-Aware Encoder (LAE), Luminance Extraction Module (LEM), and Adaptive Decoding, jointly address challenges such as object-color mismatch and color overflow.

While MLLMs offer superior semantic comprehension in complex scenes, they remain less optimized than CLIP for simpler cases and demand greater computational resources. Future work will explore lightweight MLLM integration and generalization strategies to enable real-time, resource-efficient colorization without compromising quality.

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
