# Supplementary Material for "M-Color: MLLM-Guided Diffusion Models for Image Colorization"

ICLR'25

# Additional Qualitative Results

In this supplementary material, we present additional qualitative results of our proposed method, M-Color, which demonstrates the effectiveness of MLLM-guided diffusion models in image colorization.

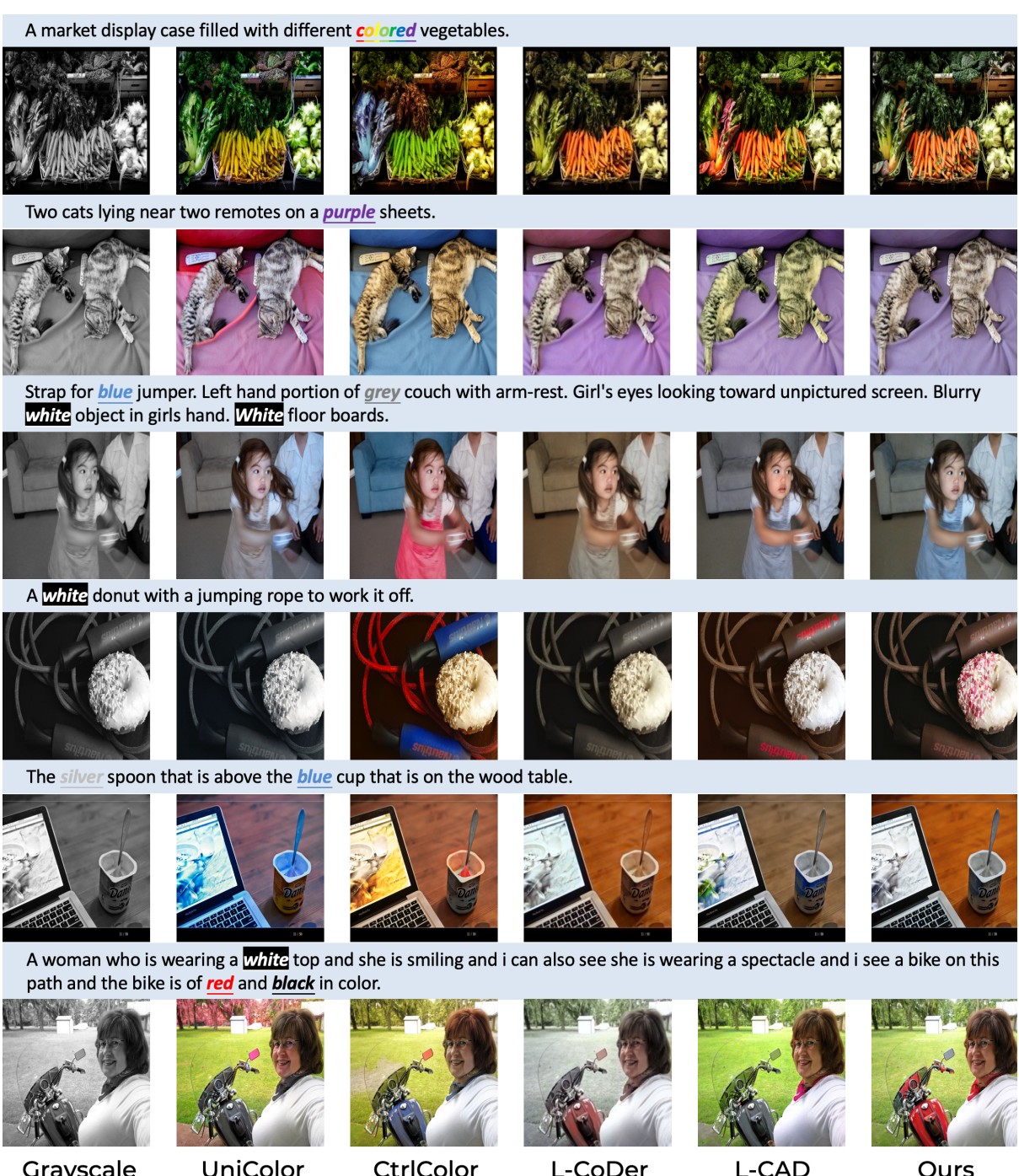

Figure 1: Additional qualitative results obtained using M-Color. These results illustrate the enhanced colorization quality achieved by our model.