# OpenReview forum: "M-COLOR: MLLM-GUIDED DIFFUSION MODELS FOR IMAGE COLORIZATION"
_ICLR.cc/2026/Conference — Submitted to ICLR 2026_

### Official Review · Reviewer_o9QQ · 2025-10-16

**Soundness:** 2
**Presentation:** 1
**Contribution:** 1
**Rating:** 2
**Confidence:** 5

**Summary:**

The paper proposes M-Color, a language-guided image colorization pipeline built on Stable Diffusion. It adds a Luminance-Aware Encoder (LAE) for structure preservation, a Luminance Extraction Module (LEM) to inject luminance features into the UNet, and an MLLM-based text embedding with “Adaptive Decoding.”

**Strengths:**

- Clear decomposition of goals: structure via LAE/LEM, semantics via MLLM embeddings.
- Includes a small user study on text matching and realism with reported wins.

**Weaknesses:**

- Outdated backbone: the core method is built on Stable Diffusion; no evidence on SDXL/modern DiT/flow models.
- LEM appears ad-hoc: described as depthwise + pointwise convs to reshape LAE features; the paper gives little principle beyond adaptation, and no swaps against simple MLPs or alternative adapters.
- Presentation issues: the first page has a stray “1”, Figure 2 is hard to parse (unclear z_t vs z_gray flows; zero_convs/LEM interactions), and Tables 1/3 are oddly formatted, which hurts clarity.

**Questions:**

- What is the theoretical role of LEM beyond channel/spatial matching? Please compare LEM with (a) an MLP adapter, (b) a 1×1 conv, under similar params/FLOPs.
- Does M-Color transfer to SDXL or a recent DiT/flow backbone without redesign? Provide results or discuss blockers.
- How much do gains persist if you replace the MLLM with CLIP or a small text encoder? Report cost–benefit (quality vs latency/VRAM)

---

### Official Review · Reviewer_cdmB · 2025-10-31

**Soundness:** 3
**Presentation:** 3
**Contribution:** 2
**Rating:** 4
**Confidence:** 4

**Summary:**

The paper introduces M-COLOR, a framework for language-guided image colorization that leverages Multimodal Large Language Models (MLLMs) and diffusion models. Its technical innovations include a luminance-aware encoder and extraction module that preserve structural details and enhance consistency, a dual-encoder design that disentangles semantic and luminance features for more effective guidance, and an adaptive decoding strategy that improves semantic alignment by optimizing visual token placement from MLLMs. The method achieves state-of-the-art performance on MSCOCO and Multi-Instance datasets across quantitative metrics (PSNR, SSIM, LPIPS) and qualitative evaluations.

**Strengths:**

1. The architecture is well-motivated and technically sound.
2. Experiments are thorough, with comparisons to multiple baselines and ablations. Quantitative results show clear improvements over prior methods.
3. The paper is generally well-written and structured. Figures and tables are informative and support the claims.
4. The work pushes the boundary of controllable image colorization with improved colorization quality.

**Weaknesses:**

1. Limited Novelty: The integration of MLLMs into colorization is not new. Prior works like L-CAD and CtrlColor have already explored this direction. The architectural changes in M-COLOR, while useful, appear incremental.
2. Modest Component Impact: Improvements from Adaptive Decoding in Table 3 are relatively small, raising questions about their standalone value. Besides, the ablation study shows that removing Adaptive Decoding leads to only minor degradation, which seems inconsistent with its visual impact in Fig. 7.
3. Lack of Failure Analysis: The paper does not discuss failure cases or limitations, such as prompt misalignment or semantic ambiguity.

**Questions:**

1. Could you clarify why Adaptive Decoding and CLIP text guidance appear impactful in Fig. 7 but show limited quantitative degradation in Table 3? Is there a synergistic effect among components?
2. Are there examples where M-COLOR fails to align with the prompt or introduces artifacts?
3. How well does M-COLOR generalize to out-of-distribution grayscale images (e.g., historical photos, medical scans)? Have you tested on real-world grayscale datasets beyond COCO?
4. What is the inference time compared to L-CAD or CtrlColor?

---

### Official Review · Reviewer_1Uqy · 2025-11-01

**Soundness:** 2
**Presentation:** 2
**Contribution:** 2
**Rating:** 4
**Confidence:** 4

**Summary:**

This paper introduces M-Color, a diffusion-based framework designed for language-guided image colorization. The authors identify a key limitation in existing methods: their reliance on standard CLIP text embeddings, which can result in suboptimal colorization due to a lack of deep semantic understanding.

**Strengths:**

1. Comprehensive ablation studies are provided, which systematically validate the significance of each proposed component and confirm the overall effectiveness of the framework's modules.

2. The manuscript is clearly articulated with a well-defined rationale, making the research easy for the audience to understand and follow.

**Weaknesses:**

1. The degree of technical novelty appears somewhat constrained. Beyond the novel use of MLLM embeddings, the Luminance-Aware Image Reconstruction component was previously introduced in L-CAD, and the Dual-Encoder Latent Generation is fundamentally a process of downsampling features from the LAE. Nonetheless, this is not considered a critical deficiency.

2. The paper's own qualitative results are unconvincing and contain significant artifacts. Figure 1, which serves as the primary visual showcase for the method, exhibits clear "color bleeding." For instance, in the "Single-Object Color Control" row, the color of the bus (e.g., pink, purple) visibly bleeds onto the surrounding windows and the road pavement. This artifact directly contradicts the paper's claims of "proficient control" (as stated in the abstract and Figure 1 caption) and "structural consistency."

3. The paper does not present any results on the colorization of authentic historical or vintage photographs. This omission leads to questions regarding the method's generalization capabilities and its utility in real-world applications.

**Questions:**

1. To better evaluate the model's claimed fine-grained control, can the authors provide results on more challenging tasks? For example, the current figures do not show if the model can assign different colors to different instances of the same object category (e.g., 'a red flower next to a yellow flower').

2. The presentation of tables in this paper is highly problematic. The tables lack standard gridlines or clear separators, which results in a visually confusing layout. This makes it extremely difficult for readers to align rows and columns, severely harming readability. This unconventional formatting deviates significantly from academic publishing standards and makes the entire paper appear highly unprofessional.

---

### Official Review · Reviewer_pdfm · 2025-11-04

**Soundness:** 2
**Presentation:** 2
**Contribution:** 1
**Rating:** 2
**Confidence:** 5

**Summary:**

the paper proposes to use MGIE text embedding to enhance text guidance for text-based image colorization

**Strengths:**

- the proposed framework is straightforward and easy to implement
- experiments demonstrate the effectiveness of the proposed method

**Weaknesses:**

- the usage of MLLM text conditions seems not well justified. for the simple prompts like color-object, the user can already use stroke-based colorization or existing text-conditioned approach to precisely assign colors to objects. the demonstrated non-color adjectives also do not show correct looking & feelings that correspond to the adjectives
- the results do not look visually pleasing. we can still observe color bleeding and uncolorized regions in the results. it's hard to tell if the proposed framework is the sota
- the proposed luminance-aware image reconstruction was already proposed in some previous works 2 years ago, but the paper still claims it as a major contribution
- the usage of stable diffusion is not well justified, considering there are many other better base models and some of them even have MLLM built-in
- the quantitative comparisons only use PSNR/SSIM/LPIPS, which are unreasonable for image colorization, considering the fact the colorization could have multiple valid/plausible solutions. FID/colorfullness should also be used. Besides, it is unclear whether/what captions are used for the quantitative comparisons
- the scale of user study is limited. the background of participants and the user study design are not mentioned

**Questions:**

please refer to weaknesses section

---

### Meta-Review · Area_Chair_aS8c · 2025-12-12

**Summary:**

This work presents M-Color, which is a diffusion-based language-guided image colorization framework that replaces CLIP-based text embeddings with multimodal large language models to achieve stronger semantic understanding. Besides, an adaptive decoding strategy improves language–vision alignment, while a luminance-aware encoder and luminance extraction module preserve structural and luminance consistency. Experiments show gains in semantic fidelity and visual quality over prior methods.

The submission received comments from four domain experts. They found that the technical details of this work are easy to follow and the reported empirical results are good. However, several important concerns were raised. First, the technical contributions are not strong. There is no clear evidence to demonstrate the technical novelty of the proposed module. Second, some empirical results are not convincing. For some visualizations, they are not very competitive, and sometimes they fail to adequately support the authors' claims. Besides, the experimental setting is not comprehensive. Third, some details about why the proposed method works are not very clear. Reviewers also raised some questions about this. It is hard to handle these concerns fully with simple modifications.

Given these issues, the AC recommends rejection. The authors are encouraged to address the mentioned concerns to enhance the clarity and overall impact of the work in future submissions.

**Reviewer Concerns:**

There is no response provided by the authors. Therefore, no concern raised by the reviewers could be addressed. The concerns are still outstanding.

**Reviewer Scores:**

As there is no response provided, reviewers are unable to participate in discussions with the authors. Their scores will most likely remain unchanged in the end.

---

### Decision · Program_Chairs · 2026-01-26

Reject